# Analysis of the Associations between Arthritis and Fall Histories in Korean Adults

**DOI:** 10.3390/ijerph18073758

**Published:** 2021-04-03

**Authors:** Jung Woo Lee, Seong Hun Kang, Hyo Geun Choi

**Affiliations:** 1Department of Orthopedic Surgery, Wonju College of Medicine, Yonsei University, Wonju-si 26426, Korea; berrybearlee@gmail.com; 2Department of Rheumatology, College of Medicine, Hallym University, Anyang-si 14068, Korea; shkang@hallym.or.kr; 3Department of Otorhinolaryngology-Head & Neck Surgery, College of Medicine, Hallym University, Anyang-si 14068, Korea; 4Hallym Data Science Laboratory, College of Medicine, Hallym University, Anyang-si 14068, Korea

**Keywords:** accidental falls, osteoarthritis, arthritis, rheumatoid, surveys and questionnaires

## Abstract

(1) Background: the purpose of the present study was to analyze the associations between arthritis and fall histories in Korean adults. (2) Methods: data from the 2015 and 2017 Korean Community Health Survey were analyzed. In total, 322,962 participants aged ≥40 years were included. The participants were divided into two different groups. First, the participants were divided into the ‘arthritis (osteoarthritis or rheumatoid arthritis) for entire life’ and ‘nonarthritis for entire life (comparison I)’ groups. Subsequently, the participants were divided into the ‘current arthritis’ and ‘noncurrent arthritis (comparison II)’ groups. Afterwards, we analyzed the prevalence odds ratios (pORs) of the fall histories of the participants using a logistic regression analysis with the 95% confidence interval (CI). The variables of income, education level, region of residence, smoking status, alcohol consumption, obesity, subjective health status, stress level, physical activity, and sleep hours were adjusted for as covariates. (3) Results: both the arthritis for entire life and current arthritis groups had higher prevalence rates of falls than the comparison I and comparison II groups, respectively (each *p* < 0.001). The pORs of falling ≥1 time and ≥2 times per year in the arthritis for the entire life group were 1.42 (95% CI = 1.38–1.46) and 1.69 (95% CI = 1.62–1.76), respectively. The adjusted pORs for falling ≥1 time and ≥2 times per year in the current arthritis group were 1.35 (95% CI = 1.31–1.39) and 1.56 (95% CI = 1.50–1.63), respectively. (4) Conclusions: previous arthritis has a significant impact on the risk of falling.

## 1. Introduction

Falls are a major cause of injury, hospitalization, and death among older people worldwide [1], and they account for the largest percentage of deaths from unintentional injuries [2]. Previous systematic reviews demonstrated that a link exists between knee osteoarthritis (OA) [3] or rheumatoid arthritis (RA) [4] and falls. Symptoms and physical impairments that are associated with lower limb OA, including joint pain and stiffness, muscle weakness, and altered sensory function, can be detrimental to balance performance and increase the likelihood of falling [5]. RA interferes with physical functioning and work productivity and affects the quality of life because of pain and stiffness [6]. Impaired function exposes RA patients to an increased risk of falling due to muscle weakness and stiff or painful joints [7,8]. However, the relationship between arthritis and falls and the risk factors are uncertain.

Although previous studies reported a fall risk in OA and RA, these studies have several limitations. In adults with arthritis, the prevalence of one fall and ≥2 falls within 12 months was 15.5% and 21.3%, respectively, which was higher than that in the controls [9]. The OA patients’ fall prevalence was high in a few previous studies [1,10,11], but these studies evaluated fall prevalence without differentiating between sex and age groups. A review article showed that gender discrepancies exist in fall risk among older individuals with OA [12]. Moreover, single falls and multiple falls and current treatment states were not evaluated separately.

Several previous studies reported a higher risk of falls in RA patients than in the control group, but adjustment for confounding factors was not evident. Additionally, most of the articles included a small number (<400) of patients [13,14,15,16], and they were based on information from participants collected before 2010 [13,14,15]. Most importantly, none of the forementioned studies stratified the groups by sex and age, and only one study included ORs data [17].

The etiology of falls is multifactorial, and it can result from a complex interaction among risk factors [3,18]. Many physical impairments that are associated with OA are known risk factors for falls [1], and falls in people with RA are correlated with factors commonly observed in older people [19]. The risk factors for falls may be modifiable or nonmodifiable and they are frequently categorized as person specific (intrinsic) or environmental (extrinsic) [3]. Personal factors may include age, sex, physical activity, sleep hours and environmental factors, including income, educational level [20], and region of residence [1,21]. However, to date, a study adjusting for confounding factors has not been published.

To analyze the associations between arthritis and fall histories, the utilization of a large, representative, and nationwide population is needed. In contrast to hospital-based studies, the present study aimed to analyze the association between arthritis (including OA and RA) and fall histories among a large representative population of Koreans who participated in a national survey. The results were adjusted to control for various confounding factors. Arthritis can differ in its etiology; OA tends to occur in the lower limbs and it is associated with joint pain, whereas RA is a chronic autoimmune disease that tends to occur in the smaller joints [22]. However, according to a recent review article, pain and physical function scores are highly significantly correlated (r > 0.7, *p* > 0.001) in both RA and OA patients [23]. To the best of our knowledge, this study is the first to demonstrate the effects of arthritis on falls in a wide range of age groups while considering numerous covariates using representative claims data.

In this study, the participants were divided into the ‘arthritis for entire life’ and ‘nonarthritis for entire life’ groups, and the participants in the ‘arthritis for entire life’ group were divided into the ‘current arthritis’ and ‘noncurrent arthritis’ groups. The ratios of single and multiple falls per year were calculated, and a crude model and a model adjusted for age, sex, income, education level, region of residence, smoking status, alcohol consumption, obesity, subjective health status, stress level, physical activity, and sleep hours were analyzed. In the subgroup analyses, we stratified the participants according to sex and age groups, and the crude and adjusted models were also analyzed.

## 2. Materials and Methods

### 2.1. Study Population and Data Collection

This cross-sectional study evaluated the association between arthritis history and falls using data from the Korean Community Health Survey (KCHS). This study was approved by the Institutional Review Board of the Korea Centers for Disease Control and Prevention (KCDC, IRB No. 2010-02CON-22-P, 2011-05CON-04-C, 2012-07CON-01-2C, 2013-06EXP-01-3C, 2014-08EXP-09-4CA, and 2016-10-01-T-A). Written informed consent was obtained from all of the participants prior to the survey. All of the KCHS data analyses were conducted in accordance with the guidelines and regulations provided by the KCDC.

We describe the details of the KCHS in the Appendix A.

Of the 456,939 total participants, we excluded the following participants from this study: participants aged under 40 years (*n* = 111,415); participants who did not provide answers to the arthritis questions (*n* = 114); participants who did not provide answers to the fall questions (*n* = 74); participants who did not respond to the sleep hours question or who slept <3 h or >12 h (*n* = 846); participants who did not respond to the weight or height questions (*n* = 17,974); and, participants who had incomplete data regarding income, education level, smoking, alcohol consumption, subjective health status, stress level, and physical activity (*n* = 3554). Ultimately, 322,962 (males = 148,417; females = 174,545) participants were included in this study (Figure 1).

### 2.2. Fall History Questionnaire

The participants were asked ‘how many times did you slip or fall in the last year?’ [1,10,19]. The participants who had a history of slips or falls ≥1 time per year were recorded as positive [24,25].

We thought that the inclusion of a slip or fall as a single event might exaggerate the outcome variables because anyone could slip or fall once by chance regardless of their physical condition. To ensure reliability, a secondary analysis was conducted using a history of slips or falls ≥2 times per year [25].

### 2.3. Arthritis History Questionnaire

The participants were asked about their history of arthritis (OA or RA) based on two questions. Arthritis was defined in two parts. In the first question, the participants were asked if they had been diagnosed with arthritis at any point during their lifetime as follows: “Have you ever been diagnosed with arthritis (osteoarthritis or rheumatoid arthritis) by your doctor?” [22,26]. In the second question, those who were diagnosed were also asked whether they were currently being treated, as follows: “Are you currently receiving treatment?”. The participants who answered ‘yes’ to the first question were defined as the ‘arthritis for entire life’ group, and the remaining participants were considered the ‘nonarthritis for entire life (comparison I)’ group. (Figure 1) If the participants answered ‘yes’ to the second question, then they were classified as the ‘current arthritis’ group, and those who answered ‘no’ were considered the ‘noncurrent arthritis (comparison II)’ group (Figure 1).

### 2.4. Other Covariates

The patients’ income, education level, region of residence, smoking status, alcohol consumption, obesity, subjective health status, stress level, physical activity, and sleep hours were specified as the variables [24,25]. The details are provided in the Appendix A.

### 2.5. Statistical Analysis

The general characteristics were compared according to the arthritis type in the entire and current arthritis groups. A two-sample t-test was used to compare the continuous variables, and a chi-square test was used to compare the categorical variables.

A crude model (simple model) and adjusted model (adjusted for age, sex, income, education level, region of residence, smoking status, alcohol consumption, obesity, subjective health status, stress level, physical activity, and sleep hours) were analyzed using multiple logistic regression to calculate the prevalence ORs (pORs) with 95% CIs of falling (≥1 time or ≥2 times per year) among the arthritis group and the entire and current arthritis groups [25].

For the subgroup analyses, we stratified the participants according to sex (male and female) and age (40 to 49 years; 50 to 59 years; 60 to 69 years; and, 70+ years). In the subgroup analyses, the pORs and 95% CIs of the crude and adjusted models were calculated using multiple logistic regression.

Two-tailed analyses were conducted, and *P*-values below 0.05 were considered to be significant. The results were statistically analyzed using SAS version 9.4 (SAS Institute Inc., Cary, NC, USA).

## 3. Results

Of the 322,962 participants, 58,847 (18.2%) participants were defined as the ‘arthritis for entire life’ group, and 264,115 (81.8%) participants were defined as the ‘nonarthritis for entire life’ group (comparison I group). Based on the second question, 33,873 (10.5%) participants were defined as the ‘current arthritis’ group, and 289,089 (89.5) participants were defined as the ‘noncurrent arthritis’ group (comparison II group). The analysis of the general characteristics showed that an older age, female sex, lower economic level, a lower education level, living in a small city, fewer days of physical activity, no smoking, no alcohol consumption, poor subjective health status, a higher stress level, and fewer sleep hours were associated with both ‘arthritis for entire life’ and ‘current arthritis’ as compared to the comparison I and comparison II groups, respectively (each *p* < 0.001). The ‘arthritis for entire life’ and ‘current arthritis’ groups had higher incidence rates of falls than the comparison I and comparison II groups, respectively (each *p* < 0.001; Table 1). The incidence of falling (≥ 1time or ≥2 times per year) was similar in the two study designs. Falling ≥1 was reported by 13,456 (22.9%) participants and falling ≥2 was reported by 5531 (9.4%) participants in the ‘arthritis for entire life’ group. Falling ≥1 was reported by 8193 (24.2%) participants, and falling ≥2 was reported by 3476 (10.3%) participants in the ‘current arthritis’ group.

The adjusted pORs of falling (≥1 time or ≥2 times per year) in the ‘arthritis for entire life’ group were 1.42 (95% CI = 1.38–1.46, *p* < 0.001) and 1.69 (95% CI = 1.62–1.76, *p* < 0.001), respectively. Those who had been diagnosed with arthritis in their lifetime had a higher probability of falling than those in the comparison group by 42% for falling ≥1 time and 69% for falling ≥2 times per year. The results of the subgroup analyses according to sex and age were consistent (Table 2). In the ‘arthritis for entire life’ group, the adjusted pOR of falling ≥1 time (1.72, 95% CI = 1.53–1.94) or ≥2 times (1.99, 95% CI = 1.68–2.36) was the highest among the women that were aged 40–49 years.

The adjusted pORs of falling (≥1 time or ≥2 times per year) in the current arthritis group were 1.35 (95% CI = 1.31–1.39, *p* < 0.001) and 1.56 (95% CI = 1.50–1.63, *p* < 0.001), respectively. Those who were being treated for arthritis had a higher probability of falling than those in the comparison group by 35% for falling ≥1 time and 56% for falling ≥2 times per year. The results of the subgroup analyses according to sex and age were consistent (Table 3). In the ‘current arthritis’ group, the adjusted pOR of falling ≥1 time (1.75, 95% CI = 1.47–2.09) or ≥2 times (2.03, 95% CI = 1.59–2.58) was the highest among the women that were aged 40–49 years.

## 4. Discussion

The purpose of the present study was to analyze the associations between arthritis and fall histories in Korean adults. We demonstrated the effects of arthritis on falls in a wide range of age groups while considering numerous covariates using representative claims data. Previous arthritis had a significant impact on the risk of falls in the present study. ‘Arthritis for entire life’ and ‘current arthritis’ were both significantly associated with falls (≥1 time or ≥2 times per year). Those who were previously diagnosed with arthritis had a higher probability of falling than those in the comparison group by 42% for falling ≥1 time and 69% for falling ≥2 times per year. Those who were being treated for arthritis had a higher probability of falling than those in the comparison group by 35% for falling ≥1 time and 56% for falling ≥2 times per year. These results are consistent with a previous report. Moreover, our data demonstrated increased pORs in all subgroups, being stratified by age and sex.

Falls were common, with an incidence of 22.9% in the ‘arthritis for entire life’ group and 24.2% in the ‘current arthritis’ group. This finding is consistent with previous reports of 15.5% to 34.8% [9,27], and it provides a basis for the use of the KCHS data to establish the fall risk. Among the arthritis patients, the fall risks that were associated with OA and RA were studied separately. In lower limb OA patients, the fall risk has been reported to be higher than that in controls (72% vs. 63%) [1], and the odds of falling increased as the number of symptomatic OA joints increased [10]. The fall prevalence in knee OA patients ranged from 23% to 63% [11]. The incidence of falls within the previous 12 months in RA patients varied, ranging from 33.6% to 52.2% [19,26]. The reported average number of falls within a year ranged from 1.6 to 2.4 [4,19,26], and the multiple fall rate ranged from 17.4% to 32.5% [8,28,29]. However, few studies analyzed falls in both the RA patients and control groups, and inconsistencies exist among these studies. Two studies found no statistically significant difference in the fall rate between the RA and control groups [15,16]. Another study showed a higher fall rate among women with RA than among the controls (54% vs. 44%), but the difference was not significant [13]. In contrast, another study found an increased risk of falls in both men and women with RA (odds ratio [OR] 1.54, 95% confidence interval [CI] 1.26–1.87, *p* < 0.001; and, OR 1.36, CI 1.19–1.556, *p* < 0.001) [17].

The adjusted pORs of falling ≥ 1 time in the ‘arthritis for entire life’ and ‘current arthritis’ groups were 1.42 and 1.35, respectively. These results are consistent with a previous report (OR 1.35) [30]. The pORs of falling in all arthritis groups were significantly higher than those in all of the comparison groups, and the pOR was the highest in the youngest age group and steadily decreased with increasing age. This finding suggests that younger arthritis patients are more prone to fall than their older counterparts, which may be due to age-related deterioration and functional failure influencing older participants and suggests that an older age may not be an important risk factor for falls [31]. An older age was significantly associated with the fear of falling (OR 1.03, CI 1.02–1.04), but not with falls in a previous study involving RA patients [31]. Older people who have memories of previously falling may be more careful because they have a fear of falling. In another study, the participants with mild OA symptoms were less likely to sustain recurrent or injurious falls when compared to the asymptomatic group [32]. In their study, the authors postulated that subjects with mild OA symptoms may be more aware of the presence of OA and they are more careful in their physical activity or perhaps restrict their activities of daily living. Early studies reported no association between age and falls in knee OA [33] and hip OA [34] patients, but other studies showed an association between older age and falls in arthritis [14], lower limb OA [10,32], and knee OA [11,35,36] patients. A higher OR of falling in younger (≤55 years) RA patients was observed in one study, and the highest prevalence of falling was observed in those that were aged 45 to 55 years (OR 1.18, CI 0.38–3.66) [29]. However, many studies did not show any significant association between age and falls in RA patients [7,8,13,19,28,31,37,38,39,40,41,42], and even suggested a correlation between an older age and falls [15,26,43,44].

The participants in the ‘arthritis for entire life’ and ‘current arthritis’ groups had a higher incidence (i.e., 9.4% and 10.3%, respectively) of multiple falls than those in the comparison groups in this study, which differs from previous studies involving OA (14.89%–30.1%) [34,35,45] and RA (19.0%–50.0%) [8,16,28,29,43,46] patients. The adjusted pORs of falling ≥ 2 times in the ‘arthritis for entire life’ and ‘current arthritis’ groups were 1.69 and 1.56, respectively. These results are consistent with a previous report (OR = 1.55–1.57) [30]. Additionally, the pORs were generally higher among those with multiple falls (≥2 times) in all of the subgroups, which suggested that arthritis patients are prone to frequent falls. In a study of knee pain and falls, mild and severe pain were associated with multiple falls (OR = 4.47 and OR =7.26, respectively) [47]. Stanmore et al. [28] revealed that multiple fallers with RA had significantly higher levels of fear of falling and pain scores than single fallers and non-fallers.

Female participants had higher pORs than male participants in all subgroups, which is consistent with previous studies investigating arthritis [26] and OA (incident rate ratio 1.12, CI 1.04–1.20, coefficients 0.18–0.26) [1,35,48]. This finding may be attributed to less bone formation and greater bone resorption [49,50,51] or a weaker quadriceps extension force [52] in females than males. Kramer et al. [53] explored the associations between thigh computed tomography-derived measures of body composition and functional outcomes in RA patients. In their study, as compared to males, females had a significantly smaller thigh muscle area (*p* < 0.001) and larger thigh fat area (*p* < 0.001). Additionally, a smaller thigh muscle area and larger thigh fat area were significantly associated with higher reported levels of disability and performance limitations (*p* < 0.001) in females than in males. Such contributors may account for the higher fall rate in female RA patients. Although most of the studies found no association between sex and falls, male fallers had a significantly higher incidence of falls per 1000 person-years (*p* < 0.0001) than female fallers in one study [54]. The female sex was a significant risk factor for falls in two studies [15,26], and more women tended to report multiple falls than men (OR = 1.35, CI = 1.02–1.79) in one study [44].

Knowledge regarding the pathways that link falls in OA patients with demographic, clinical, and physiological factors is limited. In a systematic review of 11 studies, the risk factors for falls in OA patients included impaired balance, muscle weakness, the presence of comorbidities, and an increasing number of symptomatic joints [3]. The authors suggested that pain may partially explain the increased risk of falling in knee OA patients. In contrast, there are opinions that pain and symptoms might protect against falls. People with arthritis are expected to walk more carefully because of pain, and increasing pain severity might be protective against falls [55]. People with arthritis fall less, because their activities are limited, and mild symptoms appear to be protective against falls [32]. In a longitudinal cohort study, a previous history of falling was the most significant predictor of falls in lower extremity OA patients [1]. Recently, people with lower back pain were reported to have 2.7- to 3.7-fold increased odds of recurrent falls [45]. Additionally, diabetes, visual impairment, and the use of opioids and antidepressants may increase the fall rate [32,56].

Relatively more research has been conducted to investigate the potential risk factors in RA than OA. Many potential risk factors for falls in RA patients have been proposed in previous studies. The use of certain medications [8,19,28,29,31] and multiple medications [28,29,42,43] were reported as risk factors. Additionally, a history of previous diseases [13,16,26,28] and surgery in the lower limbs [8,15,31] were also reported. Future studies need to acquire accurate information regarding medical histories because the preceding factors can have many effects on falls. Furthermore, functional states, such as the number of swollen/tender joints [8,13,19,39], pain according to the visual analog scale [28,39,40,43], fear of falling [26,28,31,37], postural control ability [7,8,16], walking time [16], and poor physical performance [7,8,29], were reported as the risk factors. It is important for researchers to use a consistent evaluation method, as it is difficult to objectively judge the function of patients. However, most of the previous studies assessed a small number (<500) of participants, and the results may not be representative of all adults with RA [16].

Falls have been widely recognized as a complex but preventable health issue among older people [57]. Fall prevention interventions may comprise single component interventions (e.g., exercise) or combinations of two or more different types of interventions. A review article identified multifactorial and multiple component interventions as effective in preventing falls and injurious falls in community-dwelling older people [58]. Multifactorial interventions may reduce the rate of falls when compared with the control, as follows: rate ratio (RR) 0.77, 95% CI 0.67–0.87; I^2^ = 88%; and, low-quality evidence. There is moderate-quality evidence suggesting that multiple interventions likely reduce the rate of falls (RR = 0.74, 95% CI = 0.60–0.91: I^2^ = 45%) and risk of falls (RR = 0.82, 95% CI = 0.74–0.90). Physical exercise has been shown to be an effective treatment for improving balance and reducing fall rates in the elderly. The meta-analysis showed improvements in dynamic balance (*p* = 0.008), static balance (*p* = 0.01), participants’ fear of falling (*p* = 0.10), balance confidence (*p* = 0.04), quality of life (*p* = 0.08), and physical performance (*p* = 0.30) in patients who underwent physical exercise when compared to controls [59].

This study has strengths and limitations. According to the definition of arthritis, there may have been diagnosis errors, because they relied on the patient’s memory. Arthritis had a lifetime prevalence of 18% in our paper, 9.6% in males and 35.6% in females. This finding is consistent with previous reports, even if compared to a study analyzed according to radiological criteria. Park et al. defined radiographic OA as Kellgren/Lawrence grade ≥2, and 9.3% of male and 28.5% of female participants were diagnosed with symptomatic OA [60]. Several papers evaluated the prevalence of arthritis while using questionnaires, and the results were similar to ours (22–25.4%) [22,61]. Additionally, the results regarding arthritis and falls using open-ended questions were similar to ours (odds 17%, 33.6%) [22,26]. In addition, KCHS surveys have been steadily conducted every year since 2008, and there is an advantage in their continuity. Although we adjusted for numerous covariates, several important confounding factors were not included. However, we demonstrated that arthritis was significantly associated with falls, regardless of the treatment state. Moreover, our data demonstrated increased pORs in all subgroups stratified by age and sex. There may be confounders related to falls in arthritis patients that we did not consider in our study. Various variables may contribute to falls in arthritis patients, such as demographic variables (marital status), health status variables (use of walking aid, low back pain, joint pain, vision impairment, number of medications, number of comorbid conditions, bone mineral density, sarcopenia, disease activity, and laboratory findings), health-related behaviors (drinking), psychological variables (fear of falling and depression), and physical performance (functional score and muscle strength) [1,7,8,10,11,13,15,16,17,19,26,28,29,33,34,36,37,38,39,40,41,42,43,45,54,55,62]. Further studies that are based on additional confounding factors and covariates using prospective study designs are warranted. In this study, arthritis patients were defined based on their answers to questions and not by diagnosis codes or diagnostic criteria. This selection method can be problematic for obtaining reliable results. However, this limitation seems to have been overcome by using two different study methods. The difference in the number of participants between the ‘arthritis for entire life’ and ‘current arthritis’ groups was 24,974, which represented 42.4% of the ‘arthritis for entire life’ group. The reliability of these data is likely acceptable despite this difference because the medication nonadherence rate was 54.1% [63] and the two-year sustained remission rate was 53.1% [64]. The present study originated from a cross-sectional, survey-based study in which an underestimation of falls due to poor recall existed [4]. The causal relationship between arthritis and falling is unclear because of the cross-sectional design. However, one of the greatest strengths of this study is the large sample size of 322,962 participants, resulting in strong statistical power. Prospective studies and prospective follow-up studies with long follow-up times are warranted [4].

## 5. Conclusions

Both arthritis for entire life and current arthritis groups had higher incidence rates of falls than the comparison groups. Both groups had a higher probability of falling than the comparison group after adjusted for age, sex, income, education level, region of residence, smoking status, alcohol consumption, obesity, subjective health status, stress level, physical activity, and sleep hours. The adjusted pORs for falling ≥1 time and ≥2 times per year were also higher than the control groups. Previous arthritis has a significant impact on the risk of falling.

## Figures and Tables

**Figure 1 ijerph-18-03758-f001:**
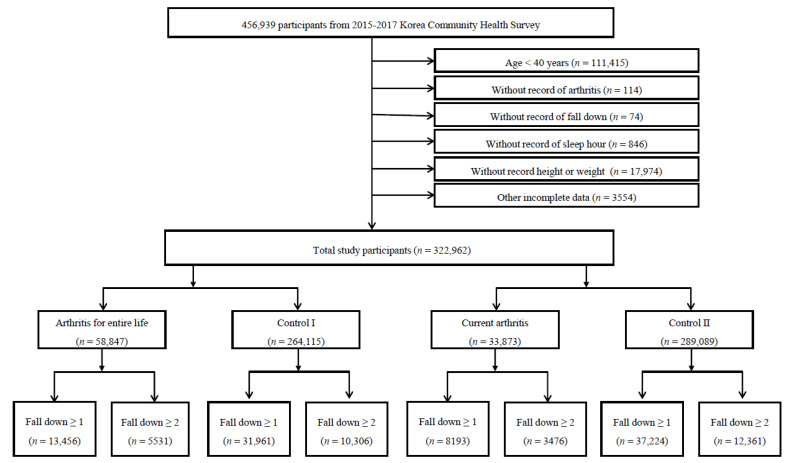
A schematic illustration of the participant selection process used in the present study. Of a total of 456,939 participants, 322,962 participants were selected. The participants were asked if they had been diagnosed with arthritis at any point during their lifetime. In the second question, those who had been diagnosed were also asked whether they were currently being treated. The participants who answered ‘yes’ to the first question were defined as the ‘arthritis for entire life’ group, and the remaining participants were considered the ‘nonarthritis for entire life (comparison I)’ group. If the participants answered ‘yes’ to the second question, then they were classified as the ‘current arthritis’ group, and those who answered ‘no’ were considered the ‘noncurrent arthritis (comparison II)’ group.

**Table 1 ijerph-18-03758-t001:** General characteristics of the participants stratified according to whether they had been diagnosed with or currently receiving treatment for arthritis.

Characteristics	Arthritis for Entire Life				Current Arthritis		
	Patient		Comparison I	*p*-Value	Patient	Comparison II	*p*-Value
Falling ≥ 1 (*n*, %)	13,456 (22.9)		31,961 (12.1)	<0.001 ^†^	8193 (24.2)	37,224 (12.9)	<0.001 ^†^
Falling ≥ 2 (*n*, %)	5531 (9.4)		10,306 (3.9)	<0.001 ^†^	3476 (10.3)	12,361 (4.3)	<0.001 ^†^
Number (*n*, %)	58,847 (18.2)		264,115 (81.8)		33,873 (10.5)	289,089 (89.5)	
Vigorous exercise/day (d, mean, SD)	0.59 (1.6)		0.83 (1.7)	<0.001 *	0.55 (1.5)	0.81 (1.7)	<0.001 *
Moderate exercise/day (d, mean, SD)	1.29 (2.2)		1.39 (2.2)	<0.001 *	1.25 (2.2)	1.39 (2.2)	<0.001 *
Age (*n*, %)				<0.001 ^†^			<0.001 ^†^
	40–49 years	2668 (4.5)	76,851 (29.1)		1063 (3.1)	78,456 (27.1)	
	50–59 years	9784 (16.6)	78,555 (29.7)		4753 (14.0)	83,586 (28.9)	
	60–69 years	18,289 (31.1)	57,612 (21.8)		10,166 (30.0)	65,735 (22.7)	
	≥70 years	28,106 (47.8)	51,097 (19.4)		17,891 (52.8)	61,312 (21.2)	
Sex (*n*, %)				<0.001 ^†^			<0.001 ^†^
	Males	13,022 (22.1)	135,395 (51.3)		6992 (20.6)	141,425 (48.9)	
	Females	45,825 (77.9)	128,720 (48.7)		26,881 (79.4)	147,664 (51.1)	
Income (*n*, %)				<0.001 ^†^			<0.001 ^†^
	Lowest	26,043 (44.3)	51,825 (19.6)		16,503 (48.7)	61,365 (21.2)	
	Low-middle	20,684 (35.2)	94,732 (35.9)		11,416 (33.7)	104,000 (36.0)	
	Upper-middle	7793 (13.2)	71,157 (26.9)		3913 (11.6)	75,037 (26.0)	
	Highest	4327 (7.4)	46,401 (17.6)		2041 (6.0)	48,687 (16.8)	
Education level (*n*, %)				<0.001 ^†^			<0.001 ^†^
	Low	44,713 (76.0)	98,668 (37.4)		27,264 (80.5)	116,117 (40.2)	
	Middle	10,005 (17.0)	93,633 (35.5)		4847 (14.3)	98,791 (34.2)	
	High	4129 (7.0)	71,814 (27.2)		1762 (5.2)	74,181 (25.7)	
Region of residence (*n*, %)							
	Urban	21,836 (37.1)	123,772 (46.9)		11,589 (34.2)	134,019 (46.4)	
	Rural	37,011 (62.9)	140,343 (53.1)		22,284 (65.8)	155,070 (53.6)	
Smoking (*n*, %)				<0.001 ^†^			<0.001 ^†^
	None	46,296 (78.7)	150,853 (57.1)		26,976 (79.6)	170,173 (58.9)	
	Past smoker	7772 (13.2)	57,791 (21.9)		4376 (12.9)	61,187 (21.2)	
	Current smoker	4779 (8.1)	55,471 (21.0)		2521 (7.4)	57,729 (20.0)	
Alcohol consumption (*n*, %)				<0.001 ^†^			<.001 ^†^
	None	31,413 (53.4)	85,486 (32.4)		19,218 (56.7)	97,681 (33.8)	
	≤1 time a month	14,233 (24.2)	62,292 (23.6)		7912 (23.4)	68,613 (23.7)	
	2–4 times a month	6101 (10.4)	50,175 (19.0)		3141 (9.3)	53,135 (18.4)	
	≥2 times a week	7100 (12.1)	66,162 (25.1)		3602 (10.6)	69,660 (24.1)	
Obesity (*n*, %)				<0.001 ^†^			<.001 ^†^
	Underweight	2756 (4.7)	9880 (3.7)		1808 (5.3)	10,828 (3.8)	
	Normal weight	22,020 (37.4)	112,818 (42.7)		12,845 (37.9)	121,993 (42.2)	
	Overweight	15,307 (26.0)	72,137 (27.3)		8648 (25.5)	78,796 (27.3)	
	Obese I	16,617 (28.2)	63,673 (24.1)		9366 (27.7)	70,924 (24.5)	
	Obese II	2147 (3.7)	5607 (2.1)		1206 (3.6)	6548 (2.3)	
Subjective health status (*n*, %)				<0.001 ^†^			<0.001 ^†^
	Good	7679 (13.1)	95,345 (36.1)		3457 (10.2)	99,567 (34.4)	
	Normal	21,374 (36.3)	118,572 (44.9)		10,847 (32.0)	129,099 (44.7)	
	Bad	29,794 (50.6)	50,198 (19.0)		19,569 (57.8)	60,423 (20.9)	
Stress (*n*, %)				<0.001 ^†^			<0.001 ^†^
	No	15,992 (27.2)	65,276 (24.7)		9222 (27.2)	72,046 (24.9)	
	Some	27,383 (46.5)	142,227 (53.9)		15,479 (45.7)	154,131 (53.3)	
	Moderate	13,239 (22.5)	49,632 (18.8)		7794 (23.0)	55,077 (19.1)	
	Severe	2233 (3.8)	6980 (2.6)		1378 (4.1)	7835 (2.7)	
Sleep (hour, *n*, %)				<0.001 ^†^			<0.001 ^†^
	≤5	16,400 (27.9)	46,570 (17.6)		9528 (28.1)	53,442 (18.5)	
	6	15,428 (26.2)	80,671 (30.5)		8633 (25.5)	87,466 (30.3)	
	7	14,766 (25.1)	84,867 (32.1)		8399 (24.8)	91,234 (31.6)	
	8	9849 (16.7)	44,046 (16.7)		5796 (17.1)	48,099 (16.6)	
	≥9	2404 (4.1)	7961 (3.0)		1517 (4.5)	8848 (3.1)	

* Two-sample t-test was performed. Significance at *p* < 0.05; ^†^ Chi-square test was performed. Significance at *p* < 0.05.

**Table 2 ijerph-18-03758-t002:** Crude and adjusted odds ratios (95% confidence interval) of falls in the ‘arthritis for entire life’ and ‘comparison I’ groups according to the sex and age groups.

Characteristics	Number	pORs of Falling (≥1 time)				pORs of Falling (≥2 times)			
		Crude	*p*-Value	Adjusted ^†^	*p*-Value	Crude	*p*-Value	Adjusted ^†^	*p*-Value
Total participants									
Arthritis for entire life	58,847	2.15 (2.11–2.20)	<0.001 *	1.42 (1.38–1.46)	<0.001 *	2.56 (2.47–2.64)	<0.001 *	1.69 (1.62–1.76)	<0.001 *
Comparison I	264,115	1		1		1		1	
Aged 40–49 years, men									
Arthritis for entire life	823	2.09 (1.75–2.50)	<0.001 *	1.69 (1.41–2.03)	<0.001 *	2.60 (2.06–3.29)	<0.001 *	1.93 (1.51–2.46)	<0.001 *
Comparison I	36,815	1		1		1		1	
Aged 50–59 years, men									
Arthritis for entire life	2018	2.07 (1.84–2.34)	<0.001 *	1.62 (1.43–1.83)	<0.001 *	2.78 (2.36–3.28)	<0.001 *	1.96 (1.65–2.33)	<0.001 *
Comparison I	38,724	1		1		1		1	
Aged 60–69 years, men									
Arthritis for entire life	3900	1.84 (1.68–2.02)	<0.001 *	1.52 (1.39–1.67)	<0.001 *	2.44 (2.13–2.80)	<0.001 *	1.81 (1.57–2.09)	<0.001 *
Comparison I	31,259	1		1		1		1	
Aged ≥ 70 years, men									
Arthritis for entire life	6281	1.69 (1.58–1.81)	<0.001 *	1.46 (1.36–1.57)	<0.001 *	2.07 (1.87–2.29)	<0.001 *	1.65 (1.49–1.83)	<0.001 *
Comparison I	28,597	1		1		1		1	
Aged 40–49 years, women									
Arthritis for entire life	1845	2.23 (1.99–2.51)	<0.001 *	1.72 (1.53–1.94)	<0.001 *	2.96 (2.52–3.48)	<0.001 *	1.99 (1.68–2.36)	<0.001 *
Comparison I	40,036	1		1		1		1	
Aged 50–59 years, women									
Arthritis for entire life	7766	1.81 (1.70–1.93)	<0.001 *	1.46 (1.37–1.56)	<0.001 *	2.42 (2.19–2.68)	<0.001 *	1.71 (1.54–1.90)	<0.001 *
Comparison I	39,831	1		1		1		1	
Aged 60–69 years, women									
Arthritis for entire life	14,389	1.56 (1.48–1.64)	<0.001 *	1.32 (1.25–1.40)	<0.001 *	2.18 (2.00–2.38)	<0.001 *	1.65 (1.50–1.81)	<0.001 *
Comparison I	26,353	1		1		1		1	
Aged ≥ 70 years, women									
Arthritis for entire life	21,825	1.54 (1.48–1.61)	<0.001 *	1.37 (1.31–1.44)	<0.001 *	1.92 (1.80–2.06)	<0.001 *	1.61 (1.50–1.73)	<0.001 *
Comparison I	22,500	1		1		1		1	

pORs = prevalence odds ratios; Each comparison group belongs to the same age group *, Logistic regressions. Significance at *p* < 0.05; ^†^ Adjusted for age, sex, income, education level, region of residence, smoking, alcohol consumption, obesity, subjective health status, stress level, physical activity, and sleep duration.

**Table 3 ijerph-18-03758-t003:** Crude and adjusted odds ratios (95% confidence interval) of falls in the ‘current arthritis’ and ‘comparison II’ groups according to sex and age groups.

Characteristics	Number	pORs of Falling (≥1 time)				pORs of Falling (≥2 times)			
		Crude	*p*-Value	Adjusted ^†^	*p*-Value	Crude	*p*-Value	Adjusted ^†^	*p*-Value
Total participants									
Current arthritis	33,873	2.16 (2.10–2.22)	<0.001 *	1.35 (1.31–1.39)	<0.001 *	2.56 (2.46–2.66)	<0.001 *	1.56 (1.50–1.63)	<0.001 *
Comparison II	289,089	1		1		1		1	
Aged 40–49 years, men									
Current arthritis	301	2.20 (1.66–2.93)	<0.001 *	1.60 (1.19–2.14)	<0.002 *	2.68 (1.84–3.91)	<0.001 *	1.70 (1.15–2.51)	<0.007 *
Comparison II	37,337	1		1		1		1	
Aged 50–59 years, men									
Current arthritis	916	2.17 (1.83–2.58)	<0.001 *	1.53 (1.28–1.83)	<0.001 *	2.84 (2.25–3.58)	<0.001 *	1.69 (1.33–2.16)	<0.001 *
Comparison II	39,826	1		1		1		1	
Aged 60–69 years, men									
Current arthritis	1986	1.92 (1.70–2.16)	<0.001 *	1.49 (1.32–1.69)	<0.001 *	2.60 (2.20–3.09)	<0.001 *	1.75 (1.46–2.09)	<0.001 *
Comparison II	33,173	1		1		1		1	
Aged ≥ 70 years, men									
Current arthritis	3789	1.76 (1.62–1.91)	<0.001 *	1.47 (1.35–1.60)	<0.001 *	2.11 (1.88–2.37)	<0.001 *	1.61 (1.43–1.82)	<0.001 *
Comparison II	31,089	1		1		1		1	
Aged 40–49 years, women									
Current arthritis	762	2.43 (2.05–2.88)	<0.001 *	1.75 (1.47–2.09)	<0.001 *	3.31 (2.63–4.17)	<0.001 *	2.03 (1.59–2.58)	<0.001 *
Comparison II	41,119	1		1		1		1	
Aged 50–59 years, women									
Current arthritis	3837	1.92 (1.77–2.08)	<0.001 *	1.49 (1.37–1.62)	<0.001 *	2.56 (2.27–2.89)	<0.001 *	1.71 (1.50–1.94)	<0.001 *
Comparison II	43,760	1		1		1		1	
Aged 60–69 years, women									
Current arthritis	8180	1.52 (1.43–1.62)	<0.001 *	1.27 (1.19–1.35)	<0.001 *	2.04 (1.85–2.24)	<0.001 *	1.50 (1.36–1.66)	<0.001 *
Comparison II	32,562	1		1		1		1	
Aged ≥ 70 years, women									
Current arthritis	14,102	1.44 (1.37–1.50)	<0.001 *	1.28 (1.22–1.34)	<0.001 *	1.74 (1.63–1.86)	<0.001 *	1.46 (1.36–1.56)	<0.001 *
Comparison II	30,223	1		1		1		1	

pOR = prevalence odd ratios. Each comparison group belongs to the same age group; * Logistic regressions. Significance at *p* < 0.05; ^†^ Adjusted for age, sex, income, education level, region of residence, smoking, alcohol consumption, obesity, subjective health status, stress level, physical activity, and sleep hours.

## Data Availability

The data included in this study are available from KCHS, but restrictions apply to availability. These data were used under a license for the current study only and are not publicly available.

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
