# Peer review of "Analysis of the Associations between Arthritis and Fall Histories in Korean Adults"

_ijerph, 2021, doi:10.3390/ijerph18073758_

Round 1

Reviewer 1 Report

Authors solved all my criticisms.

Author Response

Thank you for giving us a lot of advice on revising our journal. The revision version of our paper was also corrected by American Journal Experts.

Reviewer 2 Report

I congratulate the authors for the study, 

although it is important to cite the previous studies which have used previously some of the tools to measure the dependent variables such as the questionnaire.

In order to compare to previous studies this part is quite relevant to ensure the applicability and the validity of the study

Author Response

I congratulate the authors for the study,

although it is important to cite the previous studies which have used previously some of the tools to measure the dependent variables such as the questionnaire.

In order to compare to previous studies this part is quite relevant to ensure the applicability and the validity of the study

Answer) Thank you for your comment. As you pointed out, it seems necessary to cite references for covariates. We have found and cited the relevant content among the references we have already cited.

References of 1, 10, 19, 24, 25 were already cited for the Fall history questionnaire. We cited other references corresponding to each content in the Methods section.

Line 126: Reference (22, 26) for the Arthritis history questionnaire

Line 136: Reference (24, 25) the Other covariates

The authors have experience in papers using big data and used similar statistical methods in previous papers. We cited the Reference 25 in the Statistical analysis.

Thank you for your advice during this revision.

This manuscript is a resubmission of an earlier submission. The following is a list of the peer review reports and author responses from that submission.

Round 1

Reviewer 1 Report

The manuscript presented has some merit because of the number of patients included, however, there are many aspects that concern me with regard to its quality.

The introduction does not have a logical order and does not present ordered information focused on the object of the study. It is very dense and sometimes the reader can get lost. A well-written and argued justification for the study is not described.

An adequate introduction should have about 5 paragraphs that include the explanation of the general topic, and progress towards the specifics, it should end with the justification and objective as the final paragraph of the introduction. The authors should not use sentences like "To consider the possible effect of age, sex and other comorbidities on outcomes, we also conducted subgroup analyzes according to age and sex. "

Material and methods should begin by describing the type of study carried out.

The dependent variables studied cannot be described as "Definition of fall" or "Definition of arthritis". These terms should be modified.

Regarding the authors' consideration of "The participants who had a history of slips or falls ≥ 1 time per year were recorded as positive." I am not sure that this is a sufficiently adequate method to be able to ensure this fact. There is no bibliography to support it. I don´t understand how and why a secondary analysis is performed.

On the other hand, and with respect to the other variable, I am concerned about the validity of the answer: “Have you ever been diagnosed with arthritis (osteoarthritis or rheumatoid arthritis) by your doctor? ”

I do not consider the methods adequate or specific in accordance with the objective of the study.

I do not understand why the population is not stratified since the comparison is not possible due to the number of confusing or modifying variables such as educational level or smoking. 

Regarding the writing of the manuscript, I ask the authors to check the grammar and the agreement of verb tenses, if it is necessary to use an official translator.

Author Response

The manuscript presented has some merit because of the number of patients included, however, there are many aspects that concern me with regard to its quality.

The introduction does not have a logical order and does not present ordered information focused on the object of the study. It is very dense and sometimes the reader can get lost. A well-written and argued justification for the study is not described.

Answer) We agree with your opinion. As you pointed out, the introduction seems too long to read for readers. We added some sentences and moved some of the contents of the introduction to the discussion to make it easier to read. Thank you.

Line 41,43: added

Line 46-71: moved and modified

An adequate introduction should have about 5 paragraphs that include the explanation of the general topic, and progress towards the specifics, it should end with the justification and objective as the final paragraph of the introduction. The authors should not use sentences like "To consider the possible effect of age, sex and other comorbidities on outcomes, we also conducted subgroup analyzes according to age and sex. "

Answer) Thank you for your advice. We moved some of the contents of the introduction to the discussion and changed the number of paragraphs to five. We also removed the sentence "To consider the possible effect of age, sex and other comorbidities on outcomes, we also conducted subgroup analyzes according to age and sex.". We hope our efforts will help you read. Thank you.

Material and methods should begin by describing the type of study carried out.

Answer) Thank you for your advice. We added a sentence describing the type of study.

Line 111

This cross-sectional study evaluated the association between arthritis histories and falls using the …

The dependent variables studied cannot be described as "Definition of fall" or "Definition of arthritis". These terms should be modified.

Answer) Thank you for your advice. We changed the terms "Definition of fall" or "Definition of arthritis" to “Questionnaire for fall down history” and “Questionnaire for arthritis history”. Thank you.

Line 137, 145

Regarding the authors' consideration of "The participants who had a history of slips or falls ≥ 1 time per year were recorded as positive." I am not sure that this is a sufficiently adequate method to be able to ensure this fact. There is no bibliography to support it. I don´t understand how and why a secondary analysis is performed.

Answer) After reading, we also found out that reference is missing. We have conducted research using the same method in our previous research, and we added the papers as a reference (reference no. 27, 28). Among the previously cited papers, papers using the similar method were also added (reference no. 1, 8, 10, 13, 14, 16). In reference 1, all participants were asked to self-report if they had a fall and landed on the floor or ground in the last 12 months. In reference 10, the authors asked subjects in the last 12 months about any falls of any type. They also asked about two or more falls but did not publish the results of the analysis. In reference 14, the authors also asked about 12 months of fall and divided the fallers into 'sporadic fallers' (1–2 falls) and 'recurrent fallers' (3 falls). Since one fall down can only occur accidentally, we analyzed a secondary analysis of more than two fall downs to compensate. This analysis was for sensitivity analysis. Thank you.

On the other hand, and with respect to the other variable, I am concerned about the validity of the answer: “Have you ever been diagnosed with arthritis (osteoarthritis or rheumatoid arthritis) by your doctor? ”

Answer) As you pointed out, information bias was possible because this study was based on a questionnaire rather than medical records. Since this study was conducted is relatively recent, we think it has an advantage. In Korea, we think it is common for doctors to inform the diagnosis, and clinically, for patients to know their disease name. When receiving the questionnaire, personnel who have received professional training at the Centers for Disease Control and Prevention explained the disease and received a questionnaire after confirming that the patients understood it. You can also see a question like this in article number 12 that we cited (Byun M, Kim J, Kim M. Physical and Psychological Factors Affecting Falls in Older Patients with Arthritis. Int J Environ Res Public Health. 2020;17(3):E1098.). Thank you.

I do not consider the methods adequate or specific in accordance with the objective of the study.

Answer) We think your point makes some sense. There are many fundamental limitations in this cross-sectional study. The ability to determine definite causality was limited because our study had an observational design. And our study could not confirm the pathophysiological mechanism between arthritis and falls, as only ORs and HRs were calculated. However, we believe that viewing large-scale data itself and considering numerous covariates will be meaningful. The above is described in the section describing the limitation (Line 378-380). Thank you.

I do not understand why the population is not stratified since the comparison is not possible due to the number of confusing or modifying variables such as educational level or smoking. 

Answer) As you said, there are many modifying variables in our study. We think these variables might be able to affect the relation between arthritis and fall down in that these factors are related their health condition, and socioeconomic condition. There was a statistically significant difference between arthritis patients and comparison in variables such as education and smoking. The above is described in the results section (Line 184-189). Therefore, these variables were adjusted with multiple logistic regression. Thank you.

Regarding the writing of the manuscript, I ask the authors to check the grammar and the agreement of verb tenses, if it is necessary to use an official translator.

Answer) Even though we edited this article from English editing service form American Journal Experts using the premium service (verification code 9183-6FCD-859F-D9B4-4FCP), this might not be sufficient. Following your comments, we revised this article again. Thank you for your comments.

Reviewer 2 Report

The paper is well written, but needs some revisions, please take a look:

  1. Table 1. In the age subdivision, it appears that >70 y.o. comparison I patients fall less than >70 y.o. comparison I patients. This is unusual if you consider accidental falls. How do the authors interpret these data? 
  2. Figure 1A and figure 1B appear repetitive, try to mix then if it is possible.
  3. Lines 261-263 "This may be attributed to less bone formation ... in females than in males". Please extend this very important concept on trabecular bone and bone formation. Please consider these papers:       The Y-shaped trabecular bone structure in the odontoid process of the axis: a CT scan study in 54 healthy subjects and biomechanical considerations. J Neurosurg Spine. 2019 Feb 1:1-8. doi: 10.3171/2018.9.SPINE18396.         Bone mineral density and trabecular bone score in postmenopausal women with knee osteoarthritis and obesity. Wiad Lek. 2020;73(3):529-533. 
  4. Lines 316-317 "In our analysis, we did not include some factors related to falls in arthritis patients". What do you mean? Explain which factors or remove this sentence.
  5. Lines 277-279 "The authors suggested that pain may partly explain the increased risk of falling in knee...  mild symptoms appeared to be protective against falls". These two sentences seem contradictory. Please revise them.

Overall a good paper.

Author Response

The paper is well written, but needs some revisions, please take a look:

  1. Table 1. In the age subdivision, it appears that >70 y.o. comparison I patients fall less than >70 y.o. comparison I patients. This is unusual if you consider accidental falls. How do the authors interpret these data? 

Answer) As you pointed out, the less accidental fall in the elderly is an unusual result. In Table 2, the pOR of fall in men over 70 years old was lower than in other age groups. This can be interpreted as follows. When people become older, it may be because (1) social activity decreases, (2) they move less because they are less active at home, and (3) they are more careful because they know that if they fall, they can get seriously injured. Thank you.

  1. Figure 1A and figure 1B appear repetitive, try to mix then if it is possible.

Answer) As you said, the two figures look repeated. We mixed Figure 1A and 1B to one figure. Thank you.

  1. Lines 261-263 "This may be attributed to less bone formation ... in females than in males". Please extend this very important concept on trabecular bone and bone formation. Please consider these papers:       The Y-shaped trabecular bone structure in the odontoid process of the axis: a CT scan study in 54 healthy subjects and biomechanical considerations. J Neurosurg Spine. 2019 Feb 1:1-8. doi: 10.3171/2018.9.SPINE18396.         Bone mineral density and trabecular bone score in postmenopausal women with knee osteoarthritis and obesity. Wiad Lek. 2020;73(3):529-533. 

Answer) We reviewed and added that reference. Thank .

  1. Lines 316-317 "In our analysis, we did not include some factors related to falls in arthritis patients". What do you mean? Explain which factors or remove this sentence.

Answer) We are sorry that our expression is not accurate. We wanted to say that there are things we have not considered. The sentence was corrected as follows. Thank you.

Lines 337

There may be confounders related to falls in arthritis patients that we did not consider in our study.

  1. Lines 277-279 "The authors suggested that pain may partly explain the increased risk of falling in knee...  mild symptoms appeared to be protective against falls". These two sentences seem contradictory. Please revise them.

Answer) After reading, we also found out that it was misrepresented. We added this sentence to avoid confusion. Thank you.

Lines 298

On the contrary, there were opinions that pain and symptoms might protect against falls. People with arthritis are expected to walk more carefully because of pain and increasing pain severity might be protective against falls [55]. When people have arthritis, they fall less because their activities are limited, and mild symptoms appeared to be protective against falls [32].

Overall a good paper.

Thank you for your advice during this revision.

Paranjape & Singhania. Effect of Body Positions on Quadriceps Angle Measurement;
Rokach, A. Loneliness in Pre and Post-operative Cancer Patients: A Mini Review;
Ai, W. Participatory Action Research into Low Literates' Medical Needs in Rural Communities.

Round 2

Reviewer 1 Report

I have serious doubts about the variable 2.3 arthritis history questionnaire because it has no evidence and it has not been validated. It is necessary if you write about osteoarthritis, a diagnosis is necessary based on a classification widely established in the bibliography such as the Kellgren-Lawrence scale. If the authors have not reported any validated diagnosis method for the accuracy of the study I have to consider the publication of this paper.

Reviewer 2 Report

Authors solved all criticisms. Good job.